# A Multi-Classification Hybrid Quantum Neural Network Using an All-Qubit Multi-Observable Measurement Strategy

**DOI:** 10.3390/e24030394

**Published:** 2022-03-11

**Authors:** Yi Zeng, Hao Wang, Jin He, Qijun Huang, Sheng Chang

**Affiliations:** 1School of Physics and Technology, Wuhan University, Wuhan 430072, China; zengyi@whu.edu.cn (Y.Z.); wanghao@whu.edu.cn (H.W.); jin.he@whu.edu.cn (J.H.); huangqj@whu.edu.cn (Q.H.); 2Key Laboratory of Artificial Micro- and Nano-Structures of Ministry of Education, Wuhan University, Wuhan 430072, China

**Keywords:** hybrid quantum neural network, multi-classification, all-qubit multi-observable measurement strategy, average pooling downsampling

## Abstract

Quantum machine learning is a promising application of quantum computing for data classification. However, most of the previous research focused on binary classification, and there are few studies on multi-classification. The major challenge comes from the limitations of near-term quantum devices on the number of qubits and the size of quantum circuits. In this paper, we propose a hybrid quantum neural network to implement multi-classification of a real-world dataset. We use an average pooling downsampling strategy to reduce the dimensionality of samples, and we design a ladder-like parameterized quantum circuit to disentangle the input states. Besides this, we adopt an all-qubit multi-observable measurement strategy to capture sufficient hidden information from the quantum system. The experimental results show that our algorithm outperforms the classical neural network and performs especially well on different multi-class datasets, which provides some enlightenment for the application of quantum computing to real-world data on near-term quantum processors.

## 1. Introduction

Quantum computers can take advantage of the superposition and entanglement of quantum systems to be exponentially faster than classical computers on certain computing tasks [1]. In the present stage, the Noisy Intermediate-Scale Quantum (NISQ) [2] devices with 50–100 qubits are capable of executing circuits composed of 1000 fundamental two-qubit operations. The ‘intermediate scale’ limits the number of qubits in quantum computers, and the ‘noise’ limits the size of quantum circuits that can be executed reliably, because the noise of quantum gates will overwhelm the signal in the circuit when the number of quantum operations is large. These limitations impose a ceiling on the computational power of NISQ devices.

Machine learning is an attractive application of quantum computers, not only because it has made great progress in solving complex practical tasks on classical computers, but also because its inherent noise resistance is beneficial for realization on NISQ devices without error correction. Recently, quantum machine learning has attracted much attention, including the Quantum Autoencoder [3,4,5], Quantum Boltzmann Machine [6,7], Quantum Generative Adversarial Learning [8,9,10,11], and Quantum Kernel Method [12,13,14,15]. Besides these, there have been many studies focusing on the application of quantum machine learning in classification tasks. Quantum algorithms have inherent advantages when applied to quantum systems. H. Chen et al. [16] trained a hybrid quantum–classical neural network to distinguish two classes of quantum data. In addition, applying quantum machine learning to classify real-world data is more attractive. E. Grant et al. [17] implemented binary classification on subsets of the IRIS [18] and MINIST [19] datasets using hierarchical quantum circuits. Moreover, W. Huggins et al. [20] trained 45 pairwise classifiers based on handwritten digits 0–9 using a quantum tensor network. The quantum neural network (QNN) is also a promising method in the field of quantum classification. E. Farhi and H. Neven [21] designed a quantum neural network to classify images of handwritten digits 3 and 6. Furthermore, the hybrid quantum–classical framework has been used for data classification, where the post-processing executed by the classical processor can reduce the computational cost of the algorithm when performing some of the most challenging tasks, making the algorithm more suitable for NISQ devices. A. Skolik et al. [22] used a hybrid quantum–classical neural network trained by a layerwise learning strategy to distinguish handwritten digits 3 and 6. C. M. Wilson et al. [23] proposed an open-loop hybrid algorithm called quantum kitchen sinks (QKS) and used it to solve the binary classification problem for handwritten digits 3 and 5. As far as we know, there have been plenty of studies on binary classification using quantum machine learning, but few on multi-classification. Z. Yang and X. Zhang [24] provided a quantum deep learning scheme to classify the IRIS dataset containing three categories. R. Y. Li et al. [25] applied nested quantum annealing correction (NQAC) to train a Boltzmann machine for four-way classification on a coarse-grained version of the MNIST dataset. In a recent study, R. Wu et al. [26] proposed an end-to-end quantum machine learning scheme for eight-way classification of the MNIST dataset, which is different from the gate-based quantum circuits. Y. Li et al. [27] investigated a quantum deep convolutional neural network (QDCNN) model and employed MNIST and GTSRB datasets for 10-way classification. As they mentioned, although it is theoretically possible that the classical data can be prepared in a quantum superposition by quantum random access memory (QRAM), its physical realization will require larger quantum computers in the future. In fact, there is still a broad margin on multi-classification problems, given the limitations of accuracy and resource costs on the stage of NISQ.

Specifically, there are challenges of classifying multi-class real-world datasets on quantum processors. One of the challenges appears in how to feed the real-world data, which are vectors with many components, into quantum circuits which have a limited number of qubits in the NISQ era. To solve this problem, it is necessary to reduce the dimensionality of real-world data. Herein, we propose an average pooling downsampling strategy to compress data, since it can retain more sample features compared with normal downsampling methods.

The other challenge is how to introduce quantum nonlinearity into quantum circuits. Nonlinear operations are the key part of classical neural networks. In a quantum circuit—except for quantum measurement, which is a nonlinear operation—most quantum operations are unitary transformations that are inherently linear. To solve this problem, in our algorithm, the quantum nonlinearity is introduced by the all-qubit multi-observable measurement (AMM) strategy which calculates the expectation values of three observables on all qubits.

In this paper, we propose a hybrid quantum neural network (HQNN) to perform multi-classification tasks. The performance of the HQNN and a classical neural network (CNN) with almost the same number of parameters is compared. The algorithm is demonstrated on the MNIST [19], IRIS [18], WINE [28], and SEMEION [29] datasets. As shown in Figure 1a, in the process of dimensionality reduction, the average pooling downsampling strategy is adopted to compress the image; in the quantum part, first we use a quantum encoding circuit to prepare the initial quantum state, then we apply a parameterized quantum circuit (PQC) to perform unitary transformations on this quantum state, and, finally, we perform quantum measurements to calculate the expectation values of observables; in the classical part, we employ the classical neural network to post-process the outcomes of the quantum part, and we use a classical optimizer to adjust both the quantum and classical parameters.

## 2. Hybrid Quantum Neural Network

The architecture of the HQNN is shown in Figure 1a, and the detailed algorithm flow is summarized as follows:

The average pooling downsampling strategy is adopted to reduce the dimension of real-world data; thus, a vector $x=\left(x_{0},x_{1},\ldots ,x_{n-1}\right)$ composed of *n*-many elements is obtained.The data vector is encoded in a quantum state $|\varphi_{in}\rangle$ by applying an encoding circuit $U_{E\mathrm{nc}}={⊗}_{i=0}^{n-1}R_{y}\left(f\left(x_{i}\right)\right)$ on an initial state ${|0\rangle }^{⊗n}$, where $f\left(x_{i}\right)=\arcsin\left(x_{i}\right)$ is a nonlinear function that maps the data to angle space, and $R_{y}\left(\theta \right)|0\rangle =\cos\left(\theta /2\right)|0\rangle +\sin\left(\theta /2\right)|1\rangle$ is a rotation operation which guarantees that the amplitude of a single qubit is real.
(1)$$|\varphi_{in}\rangle =\left[\begin{array}{c} \cos\left(f\left(x_{0}\right)/2\right)\\ \sin\left(f\left(x_{0}\right)/2\right) \end{array}\right]⊗\ldots ⊗\left[\begin{array}{c} \cos\left(f\left(x_{n-1}\right)/2\right)\\ \sin\left(f\left(x_{n-1}\right)/2\right) \end{array}\right]$$ This nonlinear quantum encoding circuit can map the input data into a higher-dimensional space, which will facilitate the subsequent classification process.The parameterized quantum circuit $U_{PQC}\left(\theta \right)$ performs a series of linear transformations on the input state.The final state $U_{PQC}\left(\theta \right)U_{Enc}{|0\rangle }^{⊗n}$ is measured by calculating the expectation values of a set of observables on all qubits.The measurement outcomes are fed into a fully connected layer with the softmax activation function to generate the predicted label.The loss function between the predicted label and the true label is computed, and a classical optimizer is employed to update parameters.

This process is repeated several times until the HQNN outputs the desired result. Details of the average pooling downsampling strategy, the parameterized quantum circuit, and the all-qubit multi-observable measurement strategy are covered later in this section.

### 2.1. Average Pooling Downsampling

In this paper, we demonstrate the feasibility of our HQNN in the image classification task based on a downsampled version of the MNIST [19] dataset. MNIST is a standard classical benchmark dataset containing 60,000 training and 10,000 test samples. Each sample is a 28 by 28 pixilated grey-scale image representing handwritten digits from 0 to 9. The limitation here is that the size of the image is too large for current quantum computers.

As a traditional downsampling method, E. Farhi and H. Neven [21] resized each image down to 4 × 4 [30], and then the processed pixels were binarized to get an input string, where each component is 0 or 1. They mentioned that some images belonging to different classes originally became the same after downscaling. They then filtered the dataset to remove the ambiguous images. The disadvantage of their downscaling method is not only that much useful image information is discarded, but also that the number of effective samples is reduced due to the elimination of some contradictory processed samples. A similar preprocessing method was also used by W. Jiang et al. [31]. The digit ‘5′ is shown as an example in Figure 2a–c.

In order to alleviate this problem, we propose a strategy which can improve the representation of input data. First of all, the original image is normalized from [0, 255] to the [0, 1] range by the Min-Max scaling method. Then, the 4 pixels on each boundary which carry almost no useful information are discarded, leaving a 20 × 20 image. Next, the average value over each 5 × 5 block is computed, resulting in a 4 × 4 image. Finally, the compressed image is flattened and a vector $x=\left(x_{0},x_{1},\ldots ,x_{15}\right)$ composed of 16 elements is obtained. The result on digit ‘5′ is shown in Figure 2a,d,e. The experimental results showed that no contradictory samples appeared after the dimensionality reduction process, which indicates that the unique characteristics of each image were retained.

### 2.2. Ladder-like Parameterized Quantum Circuit

After encoding the input data into a quantum state, we apply a parameterized quantum circuit (PQC) to perform a series of linear transformations. In general, an *n*-qubit PQC can be written as:(2)$$U_{PQC}\left(\vec{\theta }\right)|\varphi_{in}\rangle =\left(\prod_{i=0}^{m-1}U_{i}\left(\theta_{i}\right)\right)|\varphi_{in}\rangle ,$$
where $U_{PQC}\left(\vec{\theta }\right)$ consists of unitary gates $U_{i}\left(\theta_{i}\right)$ with parameters $\vec{\theta }=\left\{\theta_{0},\theta_{1},\ldots ,\theta_{m-1}\right\}$, and *m* is the number of quantum gates. In the learning stage, the parameters are optimized by the classical Adam optimizer, so that the initial state can evolve to the desired state through the operation of the PQC.

Here, two-qubit unitary gates $U_{i}\left(\theta_{i}\right)=\exp\left(-i\theta_{i}Z_{j}Z_{k}\right)$ are considered as the parameterized quantum gates acting on qubit *j* and qubit *k*. It is worth mentioning that due to the fact that they are not universal gates, each one has to be decomposed into the constant elementary gate set of the physical device. As shown in Figure 1b, each $U_{i}\left(\theta_{i}\right)$ can be decomposed into two CNOT gates and one Rotation-*Z* gate with an angle of $2\theta_{i}$.

The architecture of the PQC for an *n*-dimensional input vector is illustrated in Figure 1c. We adopt a ladder-like circuit as the main part of the quantum circuit to perform the calculation. This architecture is capable of disentangling the input state as much as possible, even with a small number of quantum gates. In this ladder-like circuit, we apply two-qubit quantum gates in steps of one. In other words, a family of quantum gates with different parameters is performed on each pair of nearest-neighbor qubits, which allows us to capture the quantum correlations of a specific scale on the same layer of the network. In the terms of the input state $|\varphi_{in}\rangle$, the effective information (discrete labels in the classification task) is embedded in the quantum subsystem. Therefore, the PQC is used for performing quantum computations to extract the information hidden in this quantum subsystem. Specifically, the purpose of implementing unitary gates is to remove the superposition in quantum data, leaving the information containing the label. Then, with an appropriate measurement strategy, the classical information is extracted, but it may not be the direct representation of the label, so further classical post-processing is required.

### 2.3. All-Qubit Multi-Observable Measurement Strategy

After performing a series of unitary transformations, the input state is disentangled, and then a set of Pauli gates is applied on each qubit to extract hidden information from the final state. The statistical results of multiple observables are then passed to a classical neural network for further processing.

In most previous works, the outcome of the quantum circuit was taken from the expectation value of the Pauli *Z* operator on one [17,22,32] or two [33] readout qubits. Here, in order to adequately extract the hidden information from the disentangled quantum state and represent it as classical correlations, we calculate the expectation values of a set of Pauli operators $M=\left\{\sigma_{x},\sigma_{y},\sigma_{z}\right\}$ on all qubits, called the All-qubit Multi-observable Measurement (AMM) strategy. The quantum circuit is run several times; each time, one of the three Pauli operators is selected as the observable, and the outcomes are collected finally.

In addition, the AMM strategy is an important part of the nonlinear transformations in the proposed HQNN. In fact, realizing nonlinear transformation is important in machine learning. In a traditional neural network, a weight matrix performs linear transformation and an activation function performs nonlinear transformation. However, a quantum neural network is composed of unitary gates which perform linear transformation inherently, and only the measurement operation can introduce quantum nonlinearity. In quantum computing, the intermediate state is generally not measured, because such measurement will lead to quantum collapse and the loss of large amounts of information. To introduce more nonlinearity into a quantum neural network, we not only perform the measurement operation on all qubits, but also measure multiple observables by running the quantum circuit repeatedly. The measurement operation in our algorithm serves as the activation function of the PQC layer to achieve nonlinearity.

The expectation value of the observable *M* on the *k*th qubit is expressed as:(3)$$m_{q_{k},M}=\langle \varphi_{in}|U_{PQC}^{†}MU_{PQC}|\varphi_{in}\rangle ,$$
where $|\varphi_{in}\rangle =U_{Enc}{|0\rangle }^{⊗n}$ is the input state given by the encoding circuit, and the expectation $m_{q_{k},M}$ returns a value between −1 and 1.

An *n*-qubit circuit has a quantum output vector $\vec{m}\in R^{3n}$:(4)$${\vec{m}}_{AMM}=\left(m_{q_{0},\sigma_{x}},m_{q_{0},\sigma_{y}},m_{q_{0},\sigma_{z}},\ldots ,m_{q_{n-1},\sigma_{x}},m_{q_{n-1},\sigma_{y}},m_{q_{n-1},\sigma_{z}}\right)$$

From the two aspects of the number of measured qubits and the categories of Pauli operators, another three measurement strategies served as controls in order to make comparisons with the AMM strategy.

*Contrast* 1: The output vector is taken from the expectation value of the Pauli *Z* operator on the ancillary qubit, which has one component:(5)$${\vec{m}}_{C_{1}}=\left(m_{q_{out},\sigma_{z}}\right)$$

*Contrast* 2: The output vector is taken from the expectation values of Pauli operators $M=\left\{\sigma_{x},\sigma_{y},\sigma_{z}\right\}$ on the ancillary qubit, which has three components:(6)$${\vec{m}}_{C_{2}}=\left(m_{q_{out},\sigma_{x}},m_{q_{out},\sigma_{y}},m_{q_{out},\sigma_{z}}\right)$$

*Contrast* 3: For an *n*-qubit circuit, the output vector is taken from the expectation values of the Pauli *Z* operator on all qubits, which has *n* components:(7)$${\vec{m}}_{C_{3}}=\left(m_{q_{0},\sigma_{z}},\ldots ,m_{q_{n-1},\sigma_{z}}\right)$$

We note that *Contrast* 3 uses the same quantum circuit structure as the AMM, as shown in Figure 1c, while *Contrast* 1 and *Contrast* 2 use slightly different quantum circuits, as shown in Figure 3a, where the readout qubit is served by an ancillary qubit [21] and interacts with each data qubit at least once.

## 3. Experiments and Discussion

In this section, we present and analyze the experimental results of the proposed HQNN algorithm. First, to verify the superiority of the AMM strategy in capturing quantum information, we compare its efficiency with that of the contrast models. Next, we perform binary classification on 45 subsets of the MNIST dataset and show the comparison with some existing methods. Finally, we compare the performance of the HQNN and CNN in multi-classification tasks, and we evaluate the classification ability of the HQNN on different multi-class real-world datasets.

The experiments in this paper were implemented in the software framework TensorFlow-Quantum (TFQ) [34], which is an integration of Cirq with TensorFlow. All the parameters of the PQC were initialized as zeros. We added a fully connected layer as the classical post-processing, for which the softmax activation function for nonlinear transformation was adopted, and all the hyperparameters were set to default. In the learning stage, we used categorical cross entropy as the loss function, and we set a regularizer with *L*2 = 0.001 for the quantum part. TFQ can realize the back propagation of gradients between the quantum circuit and the classical circuit. We set the quantum circuit differentiator with the parameter shift method to calculate the gradients, and we used the Adam [35] optimizer to update parameters with a learning rate of 0.01 for binary classification and 0.02 for multi-class classification. These steps can be accomplished using standard Keras [36] tools. The binary classifiers were trained with a batch size of 256, and the multi-class classifiers were trained with the batch size of 512. The validation accuracy was recorded every epoch, and the training was stopped when validation accuracy did not increase within 30 consecutive epochs.

### 3.1. The Superiority of the AMM Strategy

In Figure 4a,b, we compare the performance of the four methods for binary and 10-way classifications of handwritten digits on the MNIST dataset, respectively. Our AMM strategy calculates the expectation values of the three Pauli operators on all qubits. For comparison, we also evaluate the other three measurement strategies in terms of the number of measured qubits and the categories of Pauli operators.

First, we performed binary classification on handwritten digits 3 and 5; the accuracies of the four methods are shown in Figure 4a. It was found that the accuracies of the three comparison models reached about 87%, while our AMM strategy increased the accuracy to 96.11%, which verifies the superiority of the AMM strategy in extracting quantum information in binary classification tasks.

Next, we verified the ability of the AMM strategy and the other three comparison models to handle multi-classification tasks. These classifiers were tasked with assigning the input sample to 1 of the 10 categories of handwritten digits 0 to 9, and their performance in the 10-way classification task is shown in Figure 4b. One can see that the AMM strategy significantly improved the classification capacity of the quantum neural network on this multi-classification task. From the perspective of the number of readout qubits, *Contrast*-2 and the AMM strategy both measure in three directions, but the former considers only one qubit as the output while AMM considers all qubits as outputs. It is obvious that our AMM strategy has greatly improved classification capacity compared with *Contrast*-2. The same is true for *Contrast*-3 compared with *Contrast*-1. This superiority is brought about by multiple measurements, the outcomes of which are correlated and therefore contain more information, whereas only one bit of information about the state of the readout qubit can be extracted from a single measurement, which is far from sufficient for complex tasks. From the perspective of measuring more observables, projecting the final state in three different directions brings a comprehensive description of the final state. For the AMM strategy, the 16-dimensional input vector is transformed into a 48-dimensional output vector. This is similar to feature mapping in machine learning, which embeds data into a higher-dimensional feature space where the data become easier to analyze. Therefore, the proposed AMM strategy is undoubtedly an excellent choice.

### 3.2. Binary Classification Based on the HQNN

In this section, we trained the HQNN to divide handwritten digits into two categories. The ladder-like PQC consisted of one-layer $U_{i}=\exp(-i\theta_{i}ZZ)$ gates, the network used the AMM strategy to perform measurement operations, and a two-way fully connected layer with the softmax activation function provided classical post-processing. The results of binary classification for each pair of digits selected from 0 to 9 are shown in Figure 5. Our HQNN was able to achieve an average accuracy above 98.46%, which indicates that the proposed network has sufficient capability to handle binary classification tasks.

Among the 45 tasks, the best performance was achieved in the task of classifying digits 1 and 4, which reached an accuracy of 99.91%. The task of classifying digits 4 and 9 performed worst overall, but it also achieved 94.53% accuracy. The reason for such a difference in classification performance may be that the digits 1 and 4 are extremely different from each other in appearance, while the difference between digits 4 and 9 is not great at all, which makes them more difficult to recognize. It is also possible that the images became blurred after dimensionality reduction, causing some representations of digits that were unlike each other initially to become similar. We believe that when quantum computers are able to provide sufficient qubits, this problem will no longer exist.

In addition, we compared the performance of our network with that of some existing methods. W. Huggins et al. [20] trained a quantum circuit consisting of 1008 parameters distributed on 63 two-qubit gates, and they showed the binary classification results for each of the 45 pairs of digits 0 through 9. Figure 6a presents a performance comparison between our network and theirs in the six tasks of classifying digits {0, 1}, {0, 4}, {1, 9}, {3, 5}, {4, 9}, and {7, 9}. Apparently, there was significant fluctuation in their work, while our network was more stable in these cases. This indicates that our network is more generalizable and has a good classification effect for most subsets, even though our network contains only 15 two-qubit quantum gates.

E. Grant et al. [17] used TTN, MERA, and a MERA pre-trained with TTN to handle four binary classification tasks. As shown in Figure 6b, we executed our network on the same four tasks. In the two simple tasks of classifying digits {0, 1} and {2, 7}, our network achieved comparable accuracy, but in the other two more complex tasks—classifying even and odd numbers and classifying numbers greater than 4 and others—the performance of our classifiers surpassed theirs. This shows that the proposed HQNN not only performs well in handling simple tasks, but also has a certain ability to deal with complex binary classification tasks.

### 3.3. Multi-Classification Based on the HQNN

In this section, we demonstrate the capability of our HQNN in multi-classification tasks. Firstly, a simple convnet from the Keras [36] documentation was used to perform 10-way classification on complete 28 × 28 images. The architecture of CNN-28 is displayed in Figure 3b. The classical neural network had more than 30,000 parameters and easily converged to 99.2% accuracy.

For a fairer comparison, we tried a simplified classical neural network to classify the downsampled 4 × 4 images. The architecture of CNN-4 is shown in Figure 3c. The CNN had 508 parameters, which is comparable to the 505 parameters contained in our HQNN. Since the difficulty increases with the number of categories, we implemented HQNN and CNN on multi-classification tasks containing 3 to 10 categories, and the results are illustrated in Figure 7a. It can be seen that, firstly, image dimensionality reduction had an adverse impact on classification performance, which will be handled by sufficient qubits in the future. Secondly, for complex compressed images, HQNN outperformed CNN, and its superiority became more obvious with increasing number of categories. For our HQNN, the accuracies when classifying images into 3 to 9 categories were all above 90%, and the 10-way classification task also achieved an accuracy of more than 89%, while only 83.33% accuracy was achieved by the CNN. The test accuracy curves of the HQNN are shown in Figure 7b.

In addition, we extracted four subsets containing five categories to verify the performance stability of our HQNN on different samples. In Figure 8a, downsampled images representing the digits 0–9 are shown, and the categories contained in each subset are marked with different colors. As mentioned above, the difference between samples affects the accuracy of classification, and the dimensionality reduction makes the images blurred and more difficult to distinguish. The accuracies of our HQNN on the four subsets fluctuated slightly, with variance lower than 0.00024. This indicates that our HQNN is universal.

Restrictions on the number of qubits and the depth of quantum circuits in NISQ devices make quantum multi-classification difficult, especially for the real-world datasets that require quantum encoding. To address this, some works choose a bypass road. M. Schuld et al. [32] simplified the multi-classification task into a set of ‘one-versus-all’ binary classification subtasks, distinguishing the digit ‘*i*’ from all other digits.

Nevertheless, there are still some works devoted to the use of quantum circuits to solve multi-classification problems, such as by W. Jiang et al. [31], who proposed a QF-Net to classify the MNIST dataset. A comparison of classification performance between QF-Net and our HQNN is shown in Figure 8b. As one can see, our HQNN performed significantly better than their classifier in both binary and multi-class classification tasks. It is worth mentioning that they used 4 × 4 downsampled images for the binary and three-way classification tasks and 8 × 8 downsampled images for the four-way and five-way classification tasks, whereas we used 4 × 4 downsampled images for all tasks. This demonstrates that, on the one hand, our average pooling downsampling strategy is conducive to the dimensionality reduction of real-world data used in quantum algorithms, which can express more information with even fewer qubits. On the other hand, our HQNN not only can handle simple multi-classification problems with a small number of categories, just like them, but also has the ability to deal with complex multi-classification problems with more categories.

Apart from this, we also verified the performance of our HQNN on other multi-class real-world datasets, as shown in Table 1. Both the IRIS [37] and WINE [38] datasets consist of three categories, with 4 attributes for each IRIS sample and 13 attributes for each WINE sample, so there is no need for dimensionality reduction. After a few epochs, both IRIS and WINE reached 100% accuracy. The SEMEION [29] dataset consists of 16 × 16 binary images with pixel values of 0 or 1 representing handwritten digits from 0 to 9. In the dimensionality reduction, the average value over each 4 × 4 block was computed, resulting in a 4 × 4 image. This 10-way classification achieved 90.98% accuracy. All in all, for datasets with small feature dimensions, dimensionality reduction is not required, and the HQNN can classify them quickly and accurately. For datasets with large feature dimensions, dimensionality reduction has a negative impact on classification performance, but our HQNN can still achieve a relatively high accuracy.

**Figure 8 entropy-24-00394-f008:**
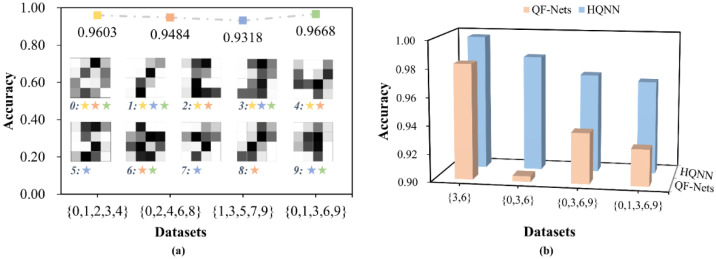
Verification of the HQNN’s performance in multi-classification tasks. (**a**) The accuracies of HQNN in four subsets of five-way classification tasks. The categories of samples contained in the four subsets and the accuracies achieved by the HQNN are marked in different colors; (**b**) A comparison of classification performance between our HQNN and the QF-Nets proposed by W. Jiang et al.

### 3.4. Computational Cost

Due to the lack of quantum resources in the NISQ era, the use of quantum circuits containing a small number of qubits and quantum gates to solve practical problems is the most appealing application of quantum algorithms. Our *n*-qubit HQNN employs (*n −* 1) two-qubit quantum gates, each of which can be decomposed into two CNOT gates and one $R_{z}$ gate, so there are 3(*n −* 1) universal quantum gates in total. The computational cost of each task is shown in Table 1, demonstrating that our HQNN has dramatically few quantum gates and can be implemented in the NISQ era.

## 4. Conclusions

In this paper, we proposed a hybrid quantum neural network to classify multi-class real-world datasets. To address the need for quantum processers that have a limited number of qubits in the NISQ era, an average pooling downsampling strategy was proposed to reduce the data dimensionality. Besides this, we adopted a ladder-like PQC with relatively few parameters as the main computing block to minimize the superposition of the input state. More importantly, all qubits in three directions are measured to convert hidden quantum information into classical correlations, which are further processed by a classical neural network. We introduced nonlinearity through three ways: injecting the nonlinear function into the encoding circuit, measuring three Pauli operations on all qubits repeatedly, and adding a softmax neural layer to the classical post-processing. The experimental results show that our HQNN not only outperforms a CNN with almost the same number of parameters, but also performs reasonably well on different multi-class real-world datasets.

However, due to the limitation of the number of qubits available in NISQ devices, some real-world datasets require dimensionality reduction. Although the average pooling downsampling strategy proposed in this paper retains more data features than traditional methods, it still loses some potentially useful information. In the near future, some algorithms, such as quantum convolutional neural networks that require fewer qubits but more computational steps, could be designed to solve this problem. This is the aim of our next work. In the long run, with the advent of large-scale quantum devices in the future, this problem will no longer exist.

With the rapid development of quantum computers, more algorithms will be designed to deal with practical problems. This multi-classification hybrid quantum neural network may provide ideas for the application of quantum algorithms in the NISQ era.

## Figures and Tables

**Figure 1 entropy-24-00394-f001:**
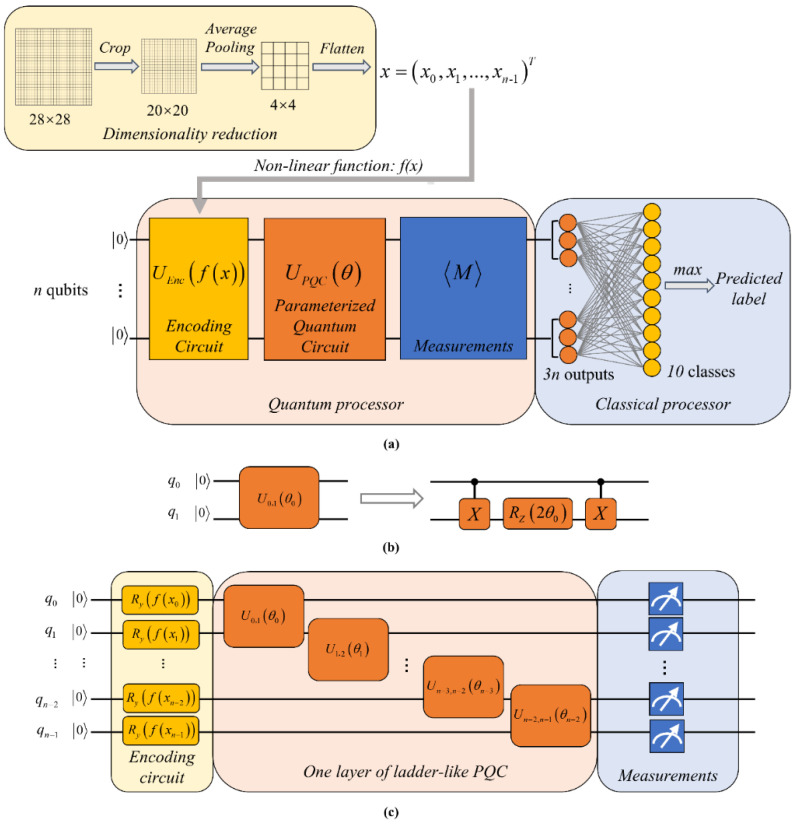
The architecture of the HQNN. (**a**) The algorithm implementation scheme; (**b**) The decomposition of two-qubit unitary gate $U_{i}\left(\theta_{i}\right)=\exp\left(-i\theta_{i}Z_{j}Z_{k}\right)$, which consists of two CNOT gates and one Rotation-*Z* gate with an angle of $2\theta_{i}$; (**c**) Details of the quantum part. The left-hand part is the encoding circuit, where $R_{y}$ is the Rotation-*Y* gate, and $f\left(x_{i}\right)=\arcsin\left(x_{i}\right)$ is the nonlinear function. The middle part is one layer of ladder-like PQC which requires (*n* − 1) parameters. The right-hand part is the measurement operation, which calculates the expectation values of a set of observables on each qubit.

**Figure 2 entropy-24-00394-f002:**
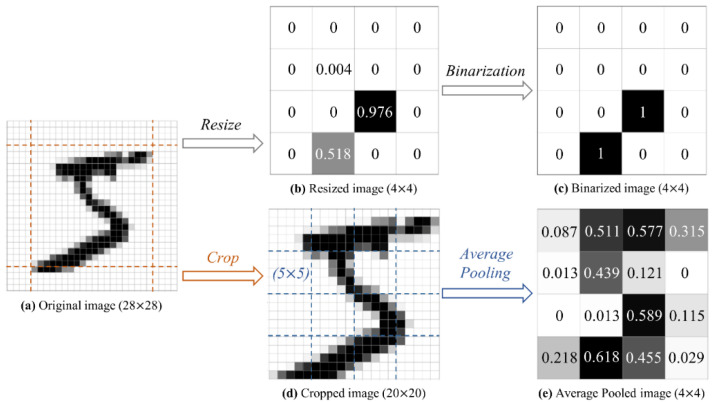
The dimensionality reduction of handwritten digit ‘5’. (**a**) The original image; (**b**) The image is reduced by the bilinear resize method. This method is a built-in function of TensorFlow and is called by the statement tf.image.resize; (**c**) The compressed image is binarized with a threshold of 0.5; (**d**) The cropped image is produced by discarding 4 pixels from each edge of the original image; (**e**) The downsampling of the image is achieved by calculating the average value of a 5 × 5 window with a stride of 5.

**Figure 3 entropy-24-00394-f003:**
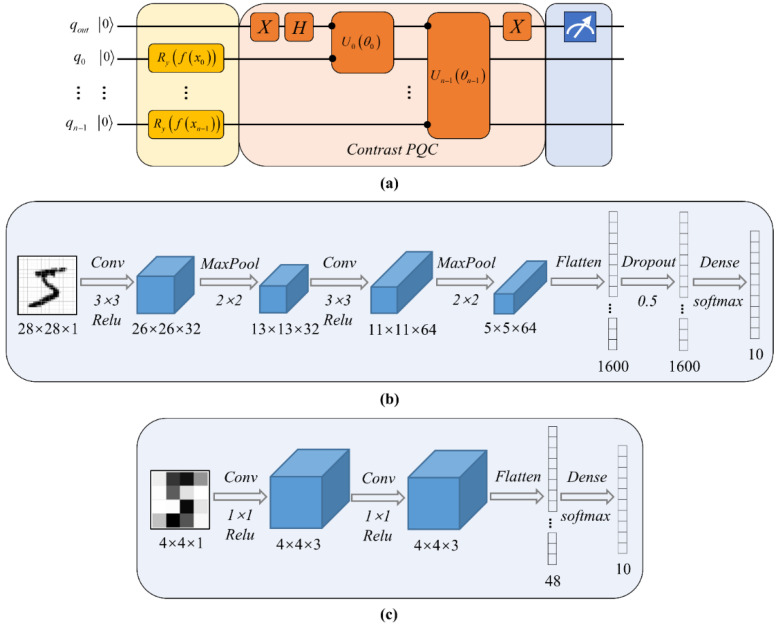
Algorithm flow charts of the contrast experiments. (**a**) The quantum circuit used by *Contrast* 1 and *Contrast* 2. The first qubit of $U_{i}\left(\theta_{i}\right)$ is the readout qubit and the second qubit is one of the other *n* data qubits; the acted qubits are marked with solid dots. Then, only the final state of the readout qubit is measured; (**b**) The architecture of CNN-28 used to classify the 28 × 28 complete images; (**c**) The architecture of CNN-4 used to classify the 4 × 4 compressed images.

**Figure 4 entropy-24-00394-f004:**
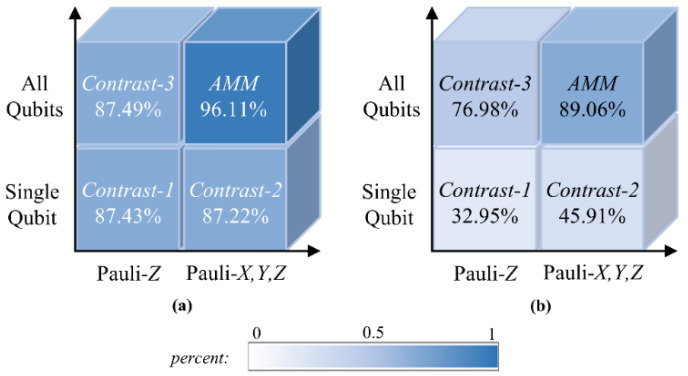
The superiority of AMM strategy and the experimental results of binary classification. (**a**) The accuracies of the four methods for binary classification on handwritten digits 3 and 5; (**b**) The accuracies of the four methods for 10-way classification on the MNIST dataset.

**Figure 5 entropy-24-00394-f005:**
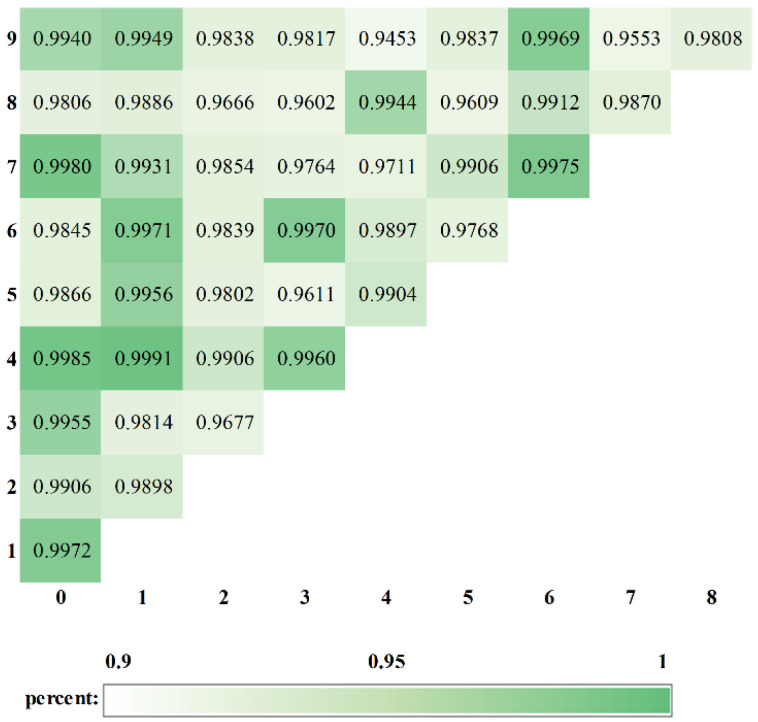
The results of binary classification on 45 subsets of the MNIST dataset.

**Figure 6 entropy-24-00394-f006:**
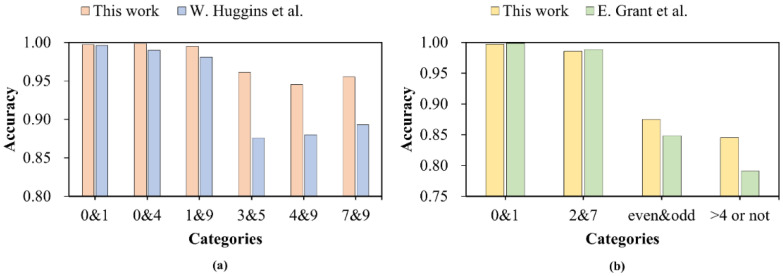
Comparison with some existing methods in binary classification tasks. (**a**) A performance comparison between our HQNN and the tensor network proposed by W. Huggins et al.; (**b**) A performance comparison between our HQNN and hierarchical quantum classifiers proposed by E. Grant et al.

**Figure 7 entropy-24-00394-f007:**
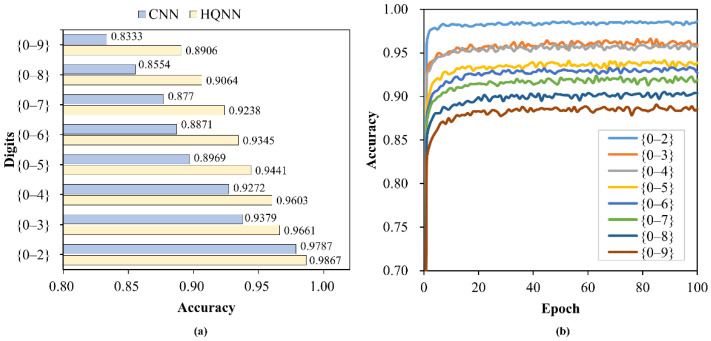
Experimental results of multi-classification tasks. (**a**) The accuracies of the HQNN and CNN in multi-classification tasks containing 3 to 10 categories; (**b**) The test accuracy curves of multi-classification tasks based on the HQNN.

**Table 1 entropy-24-00394-t001:** The classification accuracies on the different datasets.

Dataset	Dimensions	Classes	Samples		Cost		Accuracy
			Train	Test	Qubits	Gates	
IRIS	4	3	112	38	4	9	100%
WINE	13	3	133	45	13	36	100%
SEMEION	256→16	10	1194	399	16	45	90.98%
MNIST	784→16	10	60,000	10,000	16	45	89.06%

## Data Availability

All data needed to evaluate the conclusions in the paper are present in the paper and the Appendix A.

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
