# Peer review of "A Multi-Classification Hybrid Quantum Neural Network Using an All-Qubit Multi-Observable Measurement Strategy"

_entropy, 2022, doi:10.3390/e24030394_

Round 1
Reviewer 1 Report
Dear authors,
I would like to congratulate to you for this excellent work first, you give the readers a clear, careful description of the state of art, and the proposed HQNN is well explained with detail. Finally, you confirm this proposed HQNN´s pretty well performance using the experimental results.
Continue I have some questions and suggestions about the manuscript:
L.102: Is the vector X normalized? How to get the Xi?
L109: Before the line 109, eq.(1), is there a typo in the last qubit, I mean the x_0, should it be x_(n-1)?
L151-152: How to change the parameters? Some optimization method?
L153-154: How to understand the “Disentangle” and “minimize the superposition of------” physically?
P169: You mentioned that …“into classical correlations”, which possible correlations could be represented?
P307: Any reason to choose 3-10 categories?
P339: I Think it´s better using “”qubits” than “bits”.
I didn´t see any information about the resource you used to complete the experiment, it´s better introducing the tools used in this work please.
Thank you for your contribution on the development of the quantum machine learning.
Reviewer 2 Report
The manuscript is devoted to the problems concerning machine learning and quantum computing problems. As was emphasized by the Authors, the article deals with multiclassification problems, whereas most recently published articles are devoted to binary classification. They propose an application of a hybrid quantum neural network that implements a multiclassification procedure applied to real-world datasets. In the proposal considered in the manuscript, an average pooling downsampling strategy and a ladder-like parameterized quantum circuit were used. The Authors analyzed the classification accuracies and computational costs of the working model of their method, showing its advantage over the other already known methods.
The ideas and results presented in the article seem to be correct and sufficiently valid to be published. The manuscript is well written in general and is accessible to a broad range of Readers who are not necessarily experts in the field. Additionally, the Authors provide a list of references that are relevant in the studies concerning the topics discussed in the paper. Finally, one can state that the manuscript can be accepted for publication in its present form.
